

# Effects of fragrance compounds on growth of the silkworm *Bombyx mori*

Zhen-peng Kai[1], Yanwei Qiu[1], Xue-wei Zhang[1] and Shan-shan Chen[2]

[1] Shanghai Institute of Technology, Shanghai, China
[2] Shanghai Academy of Agricultural Science, Shanghai, China

## ABSTRACT

Due to the contamination and biological toxicity of some fragrance compounds, the environmental and ecological problems of such compounds have attracted more and more attention. However, studies of the toxicity of fragrance compounds for insects have been limited. The toxicity of 48 fragrance compounds for the silkworm *Bombyx mori* were investigated in this study. All of the fragrance compounds examined had no acute toxicity for *B. mori* larvae, but eight of them (menthol, maltol, musk xylene, musk tibeten, dibutyl sulfide, nerolidol, ethyl vanillin, and $\alpha$-amylcinnamaldehyde) exhibited chronic and lethal toxicity with $LC_{50}$ values from 20 to 120 $\mu$M. In a long-term feeding study, musk tibeten, nerolidol, and musk xylene showed significant growth regulatory activity. They were also extremely harmful to the cocooning of *B. mori*, resulting in small, thin, and loose cocoons. Two important insect hormones, namely, juvenile hormone (JH) and 20-hydroxyecdysone (20-E), were quantified in hemolymph following chronic exposure to musk tibeten, nerolidol, and musk xylene, respectively. Musk tibeten significantly increased JH titer and decreased the 20-E titer in hemolymph, and musk xylene had a significant inhibitory effect on JH titer and increased 20-E titer. Although nerolidol had no effect on hormone levels, exogenous JH mimic nerolidol increased the physiological effects of JH and significantly slowed the growth rate of *B. mori* larvae. The results showed that these fragrance compounds could interfere with the insect endocrine system, leading to death and abnormal growth. The risk to insects of residual fragrance compounds in the environment is worthy of attention.

## INTRODUCTION

Fragrance compounds are a group of structurally diverse additives used in a wide variety of consumer products. In recent years, there have been increasing reports of pollution of fragrance compounds in the environment and detection of them in organisms (*Weinberg, Dreyer & Ebinghaus, 2011*; *Vecchiato et al., 2018*; *Lou et al., 2016*). *Tasselli, Valenti & Guzzella (in press)* showed the current technologies are not enough efficient in removing the polycyclic musk fragrances from wastewaters in Italy. The preliminary ecological risk assessment showed that synthetic musk fragrances also pose high risk to aquatic organisms in the riverine and estuarine environment in Thailand (*Juksu et al., 2020*). HHCB-lactone was also found with the highest concentration up to 79,501 ng/g (dw) in the sludge. Low removal efficiency range from –37% (HHCB-lactone) to 58% (AHTN) were found for

Corresponding author
Shan-shan Chen,
shanshanchen2013@saas.sh.cn

four musks. In response to the contamination and biological toxicity of some fragrance compounds (such as synthetic musk fragrances), a number of studies have focused on environmental and ecological problems related to them. Synthetic musk fragrances have been detected in mussels (*Parolini et al., 2015*), fishes (*Blahova et al., 2018*), birds (*Kannan et al., 2005*), marine mammals, and sharks (*Nakata et al., 2007*; *Nakata, 2005*), as well as in human tissues and milk (*Schiavone et al., 2010*; *Moon et al., 2012*; *Kang et al., 2010*). Polycyclic musks acute toxicity to aquatic organisms ranges from hundreds of $\mu g\ L^{-1}$ to amounts of <20 mg $L^{-1}$ (*Tumová et al., 2019*). *Schreurs et al. (2004)* reported that AHTN and HHCB also have antiestrogenic effects on the human estrogen receptor in vitro. Multiple studies have shown that synthetic musk can increase cancer risk (*Zhang et al., 2017*). Musk xylene exhibits a genotoxic activity even after short exposure times following a DNA damage assay (*Rocco et al., 2015*).

Over 80% of all of the species alive on Earth are insects. They have important significance for the stable maintenance of the balance of the ecosystem (*Potts et al., 2016*). The silkworm is the main raw material source of silk and thus plays an important role in human economic life and cultural history. The silkworm is also widely used in entomological research as a model organism. Some fragrances and flavors, especially certain essential plant oils have long been reputed to repel pests (*Isman, 2000*). Despite the importance of silkworm and other insects, there have been few studies of the effects of daily chemical fragrance compounds on insects. In this study, the acute and chronic toxicity of 46 commonly used commercial fragrance compounds (including natural and synthetic compounds) were performed tested to determine their lethal toxicity and regulatory activity on the growth of *Bombyx mori*. The effects of the fragrance compounds on cocooning were also determined in a long-term feeding study. Finally, juvenile hormone (JH) and 20-hydroxyecdysone (20-E) were quantified in silkworm hemolymph to test the possible causes of toxicity.

## MATERIALS AND METHODS

### Insect
Larvae of the silkworm *B. mori* strain *p50* (DaZao) were raised from eggs provided by the Institute of Sericulture and System Biology at the Southwest University of China. The larvae were reared on the fresh mulberry leaves at 26 ± 1 °C, 70 ± 5% humidity with a photoperiod of 12 h light and 12 h dark.

### Chemicals
All fragrance compounds (Table S1) were purchased from TCI Shanghai, China. JH I, JH II, JH III and 20-E were purchased from Toronto Research Chemicals (Toronto, ON, Canada) and Sigma-Aldrich (St. Louis, MO, USA), respectively. The internal standards JH III-D3 and ponasterone A were purchased from J&K Scientific Ltd. (Beijing, China).

### Assays for to determine the impact of feeding on *B. mori* larval growth and mortality
Healthy and active larvae of similar size (larval length 3 ± 0.5 mm) at the end of the first ecdysis were randomly selected and placed in plastic containers. In the acute feeding study,

newly molted healthy larvae at the second stadium were randomly selected and fed with a small piece mulberry leaves contaminated with 2 µL fragrance solution at concentration 1 mM, and then reared on normal fresh mulberry leaves. Larvae were observed after 24 h, when dead larvae were recorded and removed. In the long-term feeding study, the larvae were exposed to the fragrance compounds from the second instar (2 µL for insects at the second instar, 4 µL at the third instar, and 6 µL at the fourth and fifth instars), administered in the same way as in the acute feeding study. Each animal was treated with 2–6 pmol day$^{-1}$ (approximately 1 ng day$^{-1}$) fragrance compounds. Each day, after the larvae had finished ingesting the contaminated mulberry leaves, fresh and clean leaves were fed to maintain their dietary requirements. The fragrance compounds were dissolved in dimethyl sulfoxide and then mixed with pure water. The concentrations ranged from 0.001 to 10 mM. The concentrations tested for each compound were shown in Table S1. Larval mortality and insect wet weight were recorded every day after treatment (*Wang et al., 1999*). Controls were treated with the solvent. There have three replications (done on separate days) at each dose and each replication contained 5–7 larvae.

## Quantitative assays of JH in hemolymph using LC-MS/MS

Newly molted fifth-instar *B. mori* (day 0) were used in this study. The hemolymph was collected in a microsyringe after cutting an abdominal leg. A volume of 50 µL hemolymph was immediately transferred to a 500 µL glass centrifuge tube containing 50 µL acetonitrile, 50 µL sodium chloride (0.9%, wt v$^{-1}$) solution, and 5 ng JH III-D3 as an internal standard. The sample was extracted twice with 100 µL hexane following vigorous vortexing and centrifuged at $2,500 \times g$ for 5 min (*Kai et al., 2018*). The hexane phase (upper layer) was removed and transferred to a new glass vial, and then dried under nitrogen flow. The residue was dissolved in one mL acetonitrile. The measurement of JHs was determined using a liquid chromatography method coupled to electrospray tandem mass spectrometry (LC-MS/MS) reported by *Ramirez et al. (2020)*.

## Quantitative assays of 20-E in whole body using UPLC-MS/MS

The hemolymph of *B. mori* larvae (4 days into the fourth instar) was collected as described above, and 50 µL hemolymph was immediately transferred into 500 µL methanol. Ponasterone A (10 ng mL$^{-1}$) was used as the internal standard in this study. The mixture was vortexed vigorously and centrifuged at $2,500 \times g$ for 10 min. The supernatant was collected in a new microtube, and the pellet was re-extracted with 500 µL methanol. The combined methanol extracts were evaporated under a stream of nitrogen. The residue was dissolved in 100 µL 80% methanol in water (*Furuta et al., 2010*). The quantitative assay for 20-E was determined using a UPLC-MS/MS system, consisting of Waters ACQUITY UPLC I-Class (Waters Corp., Milford, MA, USA), coupled with AB SCIEX TRIPLE QUAD$^{TM}$ 5,500 mass spectrometer from AB Sciex (Framingham, MA, USA).

An ACQUITY UPLC$^®$BEH C18 column (2.1 mm $\times$100 mm, 1.7 µm) from Waters was used for separation at 40 °C with a sample injection volume of 3 µL. A binary mobile phase was composed of (A) methanol and (B) 0.1% formic acid in water. A linear mobile phase gradient started at 10% A (0–1 min) and went to 90% A at 4 min (held until 7

min), followed by a gradient to 10% A at 7.1 min (held until 8 min) at a flow rate of 0.3 mL min$^{-1}$. The MS determination was performed in positive electrospray mode with monitoring of the two most abundant MS/MS (precursor/product) ion transitions using scheduled multiple-reaction monitoring (MRM) program for each analyte. The ion source conditions were as follows: ion spray voltage, 5.5 kV; block source temperature, 500 °C; ion source gas 1 pressure, 50 psi; ion source gas 2 pressure, 50 psi; curtain gas pressure, 38 psi; and collision gas (argon) pressure, 8 psi. Data acquisition and processing were carried out by Analyst software version 1.6.1 (Applied Biosystems, Foster City, CA, USA). The retention times for 20-E and ponasterone A were 3.86 min and 4.43 min, respectively. The quantification transition for 20-E was 481.3 →445.3 (declustering potential [DP]: 132 V, CE: 24 V). The confirmation transitions were 481.3 →371.2 (DP: 159 V, CE: 23 V) and 481.3 →427.5 (DP: 138 V, CE: 23 V), respectively. For the internal standard ponasterone A, the quantification transition was 465.5 →429.3 (DP: 150 V, CE: 23 V), and the confirmation transitions were 465.5 →295.3 (DP: 150 V, CE: 30 V) and 465.5 →411.2 (DP: 160 V, CE: 26 V).

## Statistics

Comparisons of the effects of fragrance compounds on larval stadium and the growth of larva were analyzed with a pooled $t$-test. The threshold for statistical significance was set at 95% ($P = 0.05$). Data were presented as percentages and were log-transformed before statistical analyses. Dose–response curves were prepared using the software GraphPad Prism version 5.0. Values are expressed as means ± standard errors (SEM), with N indicating the number of samples measured ($N = 15$–21).

## RESULTS

### Larval mortality following oral administration

A stomach toxicity test was performed to determine the larvicidal effects of the fragrance compounds in this study. After 24 h exposure following feeding at concentration 1 mM, no larvae died. This suggests that the fragrance compounds in this study have no acute toxicity for *B. mori* larvae. However, eight fragrance compounds (menthol, maltol, musk xylene, musk tibeten, dibutyl sulfide, nerolidol, ethyl vanillin, and $\alpha$-amylcinnamaldehyde) showed chronic toxicity. Most larvae that commenced feeding with these eight compounds from the second stadium died prior to pupation. The LC$_{50}$ value and mortality at 0.1 mM of each compound are shown in Table 1. These results show that the eight fragrance compounds have chronic and lethal toxicity for *B. mori* larvae. In the dead larvae, the most striking characteristic was the darkening of the cuticle in some animals and molting disturbances. This phenomenon suggested that these fragrance compounds might interfere with insect hormone levels, leading to death (*Monger et al., 1982*). No antifeedant effect was found in these experiments.

### Growth regulation

A long-term feeding study with fragrance compounds at 1 μM was performed to identify their effects on growth. We fed larvae with fragrance compounds starting at the second

**Table 1** The LC$_{50}$ values of fragrance compounds in the chronic toxicity assay.

| Compound | LC$_{50}$ value($\mu$M) (95% CI) (nonlinear fit model, df ; $R^2$) | Mortality at 0.1 mM (%) |
|---|---|---|
| Menthol | 76.6 (46.6–125.9) ($df = 25$; $R^2 = 0.95$) | $67 \pm 3$ |
| Maltol | 56.3 (39.2–80.8) ($df = 29$; $R^2 = 0.96$) | $67 \pm 3$ |
| Musk xylene | 60.7 (29.3–125.6) ($df = 21$; $R^2 = 0.92$) | $67 \pm 4$ |
| Musk tibeten | 88.1 (52.3–148.4) ($df = 27$; $R^2 = 0.97$) | $65 \pm 5$ |
| Dibutyl sulfide | 118.1 (66.2–210.7) ($df = 27$; $R^2 = 0.90$) | $51 \pm 5$ |
| Nerolidol | 63.2 (23.5–170.0) ($df = 25$; $R^2 = 0.85$) | $66 \pm 15$ |
| Ethyl vanillin | 23.8 (10.8–52.4) ($df = 23$; $R^2 = 0.95$) | $100 \pm 0$ |
| $\alpha$-amylcinnamaldehyde | 96.2 (46.5–199.0) ($df = 25$; $R^2 = 0.93$) | $55 \pm 8$ |

**Notes.**
The repeat number of each concentration was 15–21.
CI, confidence intervals; df, degrees of freedom.

instar (Day 0), and recorded the number of days from hatching to pupation (larvae that died before pupation were not recorded). Figure 1 shows musk tibeten and nerolidol significantly slowed the growth rate of *B. mori* larvae. However, musk xylene caused precocious pupation (Fig. 1). For musk tibeten, nerolidol and musk xylene, the times from newly hatched larvae to pupation were 31.6, 29.9, and 24.4 days, respectively. In the control group (not fed inhibitors), the interval was 27.4 days. Figure 2 clearly shows that larvae fed with these three fragrance compounds were significantly smaller than the control group. Four larval groups were hatched on the same day (Day 0, second instar); one was fed normal food as control, and the others were treated with 1 $\mu$M musk tibeten, nerolidol, and musk xylene, respectively. The difference in wet weight was readily apparent by comparison with the control insects at the fifth instar.

The effects of musk tibeten, nerolidol, and musk xylene on cocooning of *B. mori* were also determined with a long-term feeding study at 1 $\mu$M as above. The mean wet weight of cocoon in treatment was significantly less than in the controls (Fig. 3). Compared to the control, the wet weight of cocoon exposed to musk tibeten, nerolidol, and musk xylene lost 0.34 g cocoon$^{-1}$, 0.23 g cocoon$^{-1}$, and 0.40 g cocoon$^{-1}$, respectively. There were no significant differences between treatments and the controls on rate of cocooning.

Although mortality in the low-dose chronic exposure to fragrance compounds was minimal, the growth of *B. mori* larvae was greatly interfered with, and the mean wet weight of larvae in each larval instar were also reduced following chronic exposure to musk tibeten, nerolidol, and musk xylene. These three fragrance compounds were also extremely harmful to the cocooning of *B. mori*. The affected larvae exhibited behavioral responses, such as obstruction of cocoon-making or spinning, feeble pulse rate and final death of surviving larvae. Some larvae made only thin and loose cocoons.

### Effects on JH and 20-E titer in hemolymph by treatment with fragrance compounds

Hormones are generated and used by insects to regulate physiological, developmental, and behavioral events. Juvenile hormones and ecdysteroids are two important categories of insect hormone. Ecdysteroids are a family of steroid hormones. Their most common form

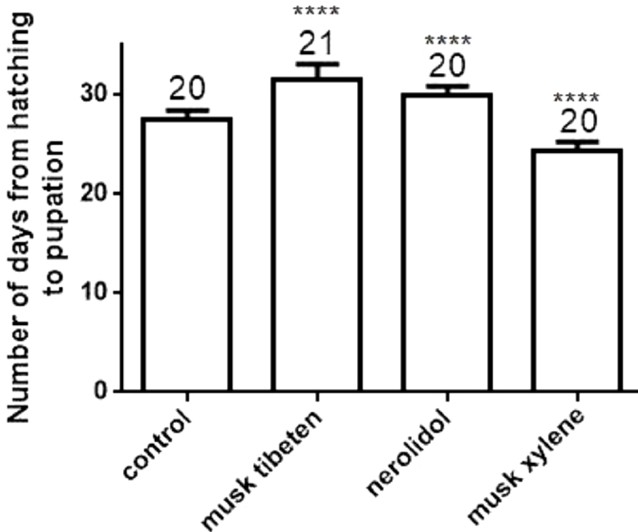

**Figure 1  Number of days from hatching to pupation following a long-term feeding study with musk tibeten, nerolidol, and musk xylene at 1 μM.** *B. mori* larvae were fed with fragrance compounds starting with the second instar, and the number of days from hatching to pupation was recorded. Values represent means ± SEMs, **** $p < 0.0001$.

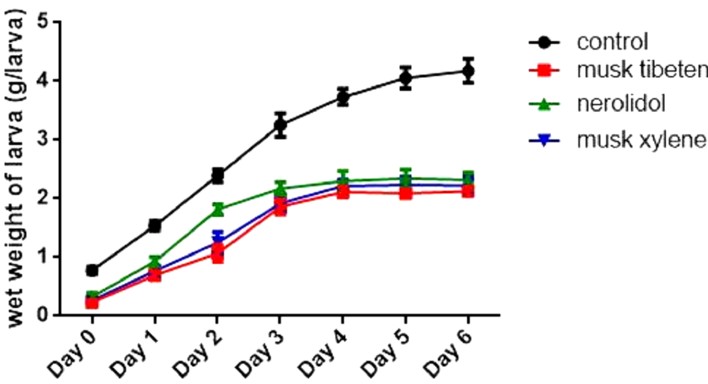

**Figure 2  Effects on larval weights of *B. mori* at the fifth instar in the chronic toxicity assay.** Four larval groups hatched on the same day (Day 0, second instar); one was fed normal food as control, and the others were treated with 1 μM musk tibeten, nerolidol, and musk xylene, respectively. Each point represents mean ± SD. $N = 20$–21 individuals.

in insects is 20-E, which is converted from ecdysone in the fat body or epidermis (*Horn et al., 1966*).

To study the possible causes of chronic toxicity, JHs (JH I, JH II and JH III) and 20-E were quantified in hemolymph following chronic exposure to musk tibeten, nerolidol, and musk xylene. Larvae were fed fragrance compounds (an approximate exposure of 1 ng day$^{-1}$), as described in Section 3.2. Figure 4 shows that musk tibeten and musk xylene had significant effects on JHs titer following treatment in vivo, while nerolidol was inactive. It is interesting that musk tibeten significantly increased JHs titer in hemolymph, whereas musk

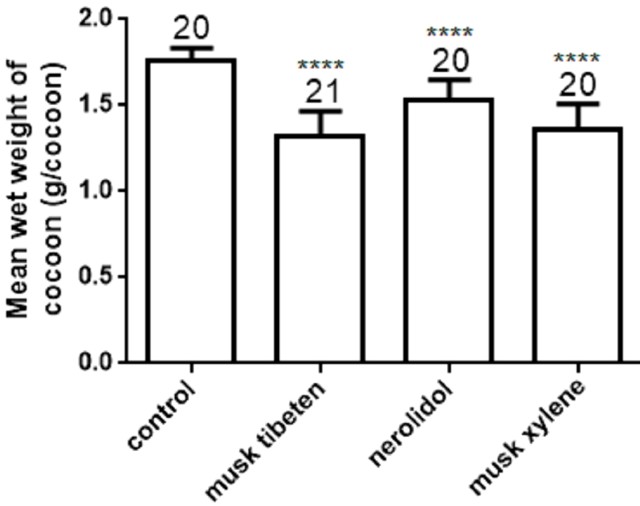

**Figure 3 Effects of musk tibeten, nerolidol, and musk xylene on the mean wet weight of cocoon in the chronic toxicity assay.** Values represent means ± SEM, **** $p < 0.0001$.

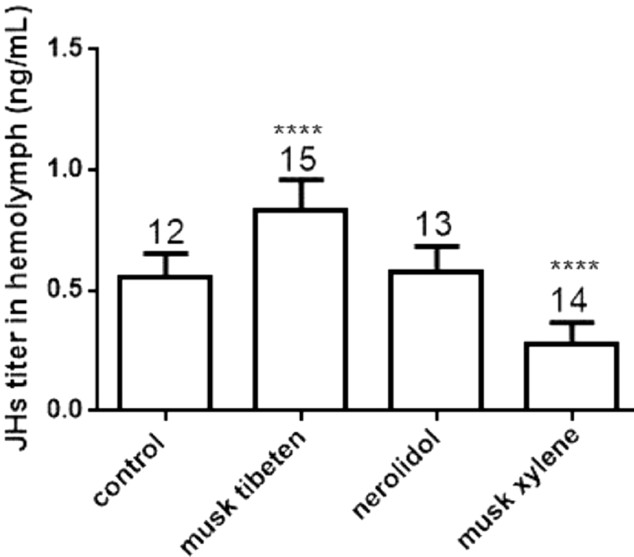

**Figure 4 Effects of musk tibeten, nerolidol, and musk xylene on hemolymph JHs titer of newly molted fifth instar *B. mori* (day 0).** Values represent means ± SEM, **** $p < 0.0001$.

xylene had significant inhibitory effect. In this study, other synthetic musk fragrances (such as musk ketone, musk AHMT, and phantolide) did not demonstrate any growth-regulatory effects on *B. mori*.

The 20-E titer in the hemolymph of *B. mori* larvae (day 4 of the fourth instar) was determined by UPLC-MS/MS (Fig. 5). Treatment with musk tibeten significantly decreased the 20-E titer in hemolymph, and musk xylene had a significantly increased effect. This result was contrary to the JH titer. Musk tibeten increased JH titer but reduced 20-E

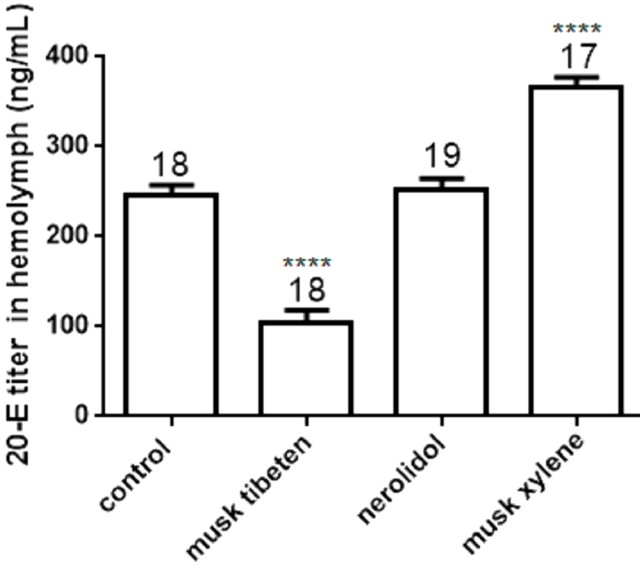

**Figure 5  Effects of musk tibeten, nerolidol, and musk xylene on hemolymph 20-E titer of _B. mori_ larvae on the fourth day of the fourth instar.** Values represent means ± SEM, **** $p < 0.0001$.

titer in hemolymph, which resulted in the retardation of the growth of _B. mori_ larvae. Must xylene had a significant inhibitory effect on JH titer but increased the 20-E titer in hemolymph. This caused precocious pupation. Nerolidol also had no effect on 20-E titer. The quantitative assays for JHs and 20-E titer indicate possible causes of growth regulation by these three fragrance compounds.

## DISCUSSION

Geraniol and geranylgeraniol have been shown to inhibit JH biosynthesis by the corpora allata in lepidopteran and blattarian insects in vitro (_Kai et al., 2018_; _Sperry & Sen, 2001_). Coumarin has been demonstrated as an ovicide inhibitor against fruit fly Drosophila melanogaster (_Shuhei & Kazuyoshi, 1980_). Toxicity tests illustrated the fumigant toxicity of 1,8-cineole against stored-grain insects (_Lee et al., 2004_). However, these four fragrance compounds above had no lethal toxicity to _B. mori_ larvae following the stomach toxicity test in this study. This suggests that different species of insects have different sensitivities to given fragrance compounds.

Our result suggests that the structural differences in fragrances have a great influence on toxicity. Musk tibeten and musk xylene are aromatic nitro compounds, and musk ketone has an aldehyde group in addition to the nitro group, however musk AHMT and phantolide do not contain nitro groups. Nerolidol is a mimic of JH, which has a similar function, but it has no effect on JH biosynthesis and metabolism (_Burdette, 1964_). In hemolymph, exogenous JH mimic nerolidol increases the physiological effects of JH and significantly slows the growth of _B. mori_ larvae.

In previous studies, some fragrance compounds have been demonstrated to have contact, fumigant, antifeedant, attractant, and repellent insecticidal actions against specific pests

(*Isman, 2000*). Due to the extensive use of fragrance compounds and the growth of the fragrance industry, fragrance residues can be detected in the environment. Concentrations of fragrance compounds in wastewater were found in concentration at $\mu g\,L^{-1}$ in a study in Italy (*Tasselli, Valenti & Guzzella, in press*). In the long-term feeding study with fragrance compounds at 1 $\mu M$ was performed to identify their effects on the growth of insect. Residual fragrance compounds may cause damage to the sericulture industry and may be harmful to other insects in the environment. The current leakage concentration may not immediately affect the ecology of insects. However, there are concerns about the deterioration of environmental pollution in the future.

## CONCLUSIONS

Although the fragrance compounds in this study have no acute toxicity for *B. mori* larvae, some exhibit chronic and lethal toxicity. In a long-term feeding study, musk tibeten, nerolidol, and musk xylene have significant growth-regulatory activity with an approximate exposure of 1 ng day$^{-1}$. They were also extremely harmful to the cocooning of *B. mori*. Larval exposure to these fragrance compounds produced small, thin, and loose cocoons. The quantitative assays for JH and 20-E titer in hemolymph showed that these fragrance compounds could interfere with hormone levels in insects, leading to death and abnormal growth. The risk to insects of residual fragrance compounds in the environment is worthy of attention.

### Funding
This work was supported by grant from the National Natural Science Foundation of China (No. 21877126). The funders had no role in study design, data collection and analysis, decision to publish, or preparation of the manuscript.

### Grant Disclosures
The following grant information was disclosed by the authors:
National Natural Science Foundation of China: 21877126.

### Competing Interests
The authors declare there are no competing interests.

### Author Contributions

- Zhen-peng Kai conceived and designed the experiments, performed the experiments, analyzed the data, prepared figures and/or tables, authored or reviewed drafts of the paper, and approved the final draft.
- Yanwei Qiu and Xue-wei Zhang performed the experiments, prepared figures and/or tables, and approved the final draft.
- Shan-shan Chen conceived and designed the experiments, performed the experiments, prepared figures and/or tables, authored or reviewed drafts of the paper, and approved the final draft.

## Data Availability

The raw measurements are available in the article and in the Supplemental Files.

## Supplemental Information

Supplemental information for this article can be found online at http://dx.doi.org/10.7717/peerj.11620#supplemental-information.

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
