# Peer review of "Effects of fragrance compounds on growth of the silkworm Bombyx mori"

_PeerJ, doi:10.7717/peerj.11620_

## Round 0.1 · original submission · Major Revisions

Apologies for the delay in getting your manuscript reviewed. The reviewer comments are straightforward. Among them are concerns about the quality of photos from which growth inhibition was concluded. The resolution of the photos is low and there are no scale bars provided. This makes it difficult to verify the growth effects. Further, there are no measurements of hormone levels in the larvae. Please address these and the other reviewer comments in your revision.

Reviewer 1 ·

Basic reporting

This paper from Kai et al is clearly written well.

Authors are investigating the effects of fragrance molecules leaked into the environment on the growth of insects.

About introduction and background, I have one fundamental question about the influence of the fragrances against insects. There are two possible effects of fragrance. The first is that the leaked fragrance molecules disperse in the atmosphere and affect the activity of insects as odors, and as a result, affect the growth of them. The other is that fragrances are accumulated in the diet of each insect such as mulberry for silkworms, and exposure by ingesting fragrance molecules with food affects growth of insects. In this paper, the authors seem to focus on the latter, but when targeting fragrances, it is preferable to also consider and introduce the former's olfactory effects in introduction section.

・In Figure 2, the Musk tibeten sample which is the most difference appears to have a lower resolution than the other photos. In addition to the image like Fig. 2, I think that it will be easier to understand if you describe the difference in growth by comparing weight and body length of larvae. It looks fine in the other figures.

・At line 206, “Musk tibeten, musk xylene, and all of the other aromatic nitro compounds (except musk ketone) do not contain nitro groups,”
I guess “these compounds contain nitro groups” is the right sentence. Authors should check.

Raw data was suppled correctly.

Experimental design

I had the impression that the standard for the amount of fragrance concentration used in both of the acute and chronic conditions was not clear throughout the experiment.

・About feeding experiments, author described that they fed with fragrance solution. The authors describe the exposure amount of Fragrance to silkworms in terms of concentration (ex 1uM), but the effects of Fragrance should also be affect by actual intake. Therefore, the authors should describe in more detail how to feed.

・Also, did the fed of Fragrance change the behavior of silkworms? If silkworm showed repellent reaction against Fragrance and the amount of food is reduced, it should be possible that there is a difference in growth and cocooning simply due to malnutrition.

・About “Larvae that died before pupation were not recorded.” at Line 116,
Isn't the proportion of larvae that died by fragrance exposure an important factor in this study titled the effects of Fragrance? Is there a difference in the number of Larvae that died before pupation for the control under Fragrance exposure conditions?

・For the effects by Nerolidol, the hormone titer was not different from the control, but the data show that the growth analysis in Fig. 1 and Fig. 3 showed a significant difference from the control. Did Neolidol cause this change by another mechanism different from hormonal pathway? And was this differences an unexpected event? I think it would be good to discuss this consideration in the discussion.

Validity of the findings

The information on fragrance concentration in wastewater mentioned in the conclusion part is quite old as 20 years ago. If you have the current information, I think it would be more realistic to raise your concerns.

Additional comments

It is interesting that this paper examines the effects of fragrance exposure using silkworms as a model insects. There are some questions described below, so I would like you to deepen the discussion. Comparing the concentration used in the experiment with the actual leakage concentration described in the conclusion section, the current leakage concentration may not immediately affect the ecology of insects. However, there are concerns about the deterioration of environmental pollution in the future.

The fragrance actually used is generally a mixture of a plurality of compounds. Therefore, it is hoped that this discovery will be the basis for future developments in fragrance mixtures and their effects on other insects.

Reviewer 2 ·

Basic reporting

The authors showed growth inhibition of compounds by the photos of larvae which were fed compounds. But resolution of the photos was very low. The photos were without scale bars. It is hard to check the growth effects.

Experimental design

I wonder the authors choose the insect of which the position in the food chain is low and the most habitats are on ground. The authors did not mention about accumulation of fragrance compounds in soils and plants. Therefore, the purpose of this MS is not very clear. Moreover, the authors did not mention the reason for select these compounds which were tested in the MS.
The author measured JH II levels of day 0 5th instar larvae. But JH I is predominant in the silkworm (Niimi and Sakurai, 1998). The should measure JH I, too.

Validity of the findings

The authors concluded that the effect of compounds on JH and 20E levels lead to death. But they did not measure the hormone levels in the larvae which treated with high dose to lead death. The photos of dead larvae is also necessary to show the effect of the compounds. They measure the hormone levels at only one developmental stage. It is not sufficient to discuss the effect of compounds on hormone level to lead the growth rate.

Annotated reviews are not available for download in order to protect the identity of reviewers who chose to remain anonymous.

Reviewer 3 ·

Basic reporting

The manuscript investigates “Effects of Fragrance Compounds on Growth of the Silkworm Bombyx mori”. This information is of interest for ecologists who work to protect the environment and entomologists also study urban pests, forest pests, agricultural pests, and merits publication. The experimental setup of this study appears to bewell-designed and the data collected carefully. However, information about the statistical data (P-values, Mean’s t-test comparisons) obtained in the treatments must be provided and detailed in the results section. I think that this manuscript requires rewriting to make its results clearer and more readily interpretable to the reader.

Experimental design

Line 93: Indicate clearly in detail how the steps were taken to solubilize the chemical compounds, for example, how much dimethyl sulfoxide was added, it was necessary to stir or heat to solubilize the compounds used in the study.

Line 176: Clarify which compounds do not show lethal toxicity to B. mori larvae in this study.

Validity of the findings

My specific comments are listed below:
1.  Apart from Vecchiato  (2018), Zhang (2017) Potts (2016) and Rocco ( 2015),  your most recent reference is from 2012. Please cite more current literature from the past five years (2015 to 2020).
2. Line 44: check and confirm; is it 2000, 2018, 2009 or 2000, 2009, 2018... Simonich; Vecchiato; Schiavone...
3. Line 165: check and confirm; Is it abbreviation stands for Lethal toxicity IC50 or LC50.
5. Improve the discussion from the conceptual point of view for acute toxicity.
6. Line 224: Was the experiment carried out just once or was it repeated?
7. Your conclusion needs to be more concise. I suggest moving lines 250-258 for the discussion.

Additional comments

This is good work and an important topic.
The science is good.

---

## Round 0.2 · Minor Revisions

Reviewer 2 has a couple of additional minor points that I would like you to address.

Reviewer 1 ·

Basic reporting

In the Revision edition, the authors have described in detail what I was concerned about in the first edition. As a result, I think it became easier to understand the significance of this paper.

There seem to be some typos as below, so please check the entire text again.
L70 favor -> flavor
L172 The results -> These results

Experimental design

no comment

Validity of the findings

no comment

Reviewer 2 ·

Basic reporting

I have read the revised MS. The MS is nearly acceptable for publication. I have only a remaining concern.
In the response, the authors stated that the measurement of hormones is to explain the effect on growth. But in line 187-188, they hypothesized that hormone levels which affected by some fragrances was leading death. The sentence in line 187-188 make readers to confuse. I think the authors should change it to an appropriate expression.

Experimental design

In Figure 2, it showed that the effect of the fragrances on the growth from day 0 to day 6. The authors mentioned that “Day 0, second instar”. I guess “day 0” in figure2 is day 0 5th instar. Is it correct?

Validity of the findings

I have some comments blow.
Line 26: Please add “(JH)” and “(20-E)” following juvenile hormone and 20-hydroxyecdysone, respectively.

Line 87: Please add space between 24 and h.

Line 91: “Each animal was treated eith …” Is it “with”?

Line 96: What is insect growth? Weight? Body length? Please specify.

Line 360: In figure 4, it is not found “***”. Please remove from figure legend.

Additional comments

This revised MS is well written.
I did not find any serious problem in the MS.

Reviewer 3 ·

Basic reporting

All requests are answered.

Experimental design

All requests are answered.

Validity of the findings

All requests are answered.

Additional comments

This is good work and an important topic. The science is good.

---

## Round 0.3 · accepted · Accept

Thank you for improving your manuscript.